# Osteoblast Secretome Modulated by Abiraterone Treatment Affects Castration Resistant Prostate Cancer Cell Proliferation

**DOI:** 10.3390/biomedicines10092154

**Published:** 2022-09-01

**Authors:** Michele Iuliani, Sonia Simonetti, Giulia Ribelli, Silvia Cavaliere, Bruno Vincenzi, Giuseppe Tonini, Francesco Pantano, Daniele Santini

**Affiliations:** 1Department of Medical Oncology, Campus Bio-Medico University of Rome, 00128 Rome, Italy; 2UOC Oncologia Universitaria, Sapienza University of Rome Polo Pontino, 04100 Latina, Italy

**Keywords:** abiraterone, castration-resistant prostate cancer (CRPC) cells, osteoblasts, androgen receptor

## Abstract

Abiraterone is a selective inhibitor of androgen biosynthesis approved for the treatment of metastatic patients affected by castration-resistant or castration-sensitive prostate cancer. Intriguingly, clinical data revealed that abiraterone also delayed disease progression in bone improving bone-related endpoints. Our group has previously demonstrated in vitro a direct effect of abiraterone on osteoclast and osteoblast function suggesting its ability to modulate bone microenvironment. Here, we performed an extensive proteomic analysis to investigate how abiraterone influences osteoblast cell secretome and, consequently, osteoblast/prostate cancer cells interaction. A panel of 507 soluble molecules were analyzed in osteoblast conditioned media (OCM) obtained from osteoblast treated or not with abiraterone. Subsequently, OCM was added to prostate cancer cells to investigate its potential effect on prostate cancer cell proliferation and androgen receptor (AR) activation status. Out of 507 screened molecules, 39 of them were differentially expressed in OCM from osteoblasts treated with abiraterone (OCM ABI) compared to OCM obtained from untreated OBs (OCM CTRL). Pathway enrichment analysis revealed that abiraterone down-modulated the release of specific osteoblast soluble factors, positively associated with cell proliferation pathways (false discovery rate adjusted *p*-value = 0.0019). In vitro validation data showed that OCM ABI treatment significantly reduced cancer proliferation in C4-2B cells (*p* = 0.022), but not in AR- negative PC-3 cells. Moreover, we also found a reduction in AR activation in C4-2B cells (*p* = 0.017) confirming the “indirect” anti-tumor AR-dependent effect of abiraterone mediated by osteoblasts. This study provides the first evidence of an additional antitumor effect of abiraterone through the modulation of multiple osteoblast proliferative signals.

## 1. Introduction

Abiraterone (ABI) is a potent, selective inhibitor of androgen biosynthesis by blocking cytochrome P450 c17 (CYP17) and suppressing testosterone production by testes, adrenals and tumor cells [1,2]. ABI demonstrated survival benefit initially in metastatic castration-resistant prostate cancer (mCRPC) patients pre-treated [3] or not with chemotherapy agents [4,5,6] and, more recently, in high-risk de novo metastatic hormone sensitive prostate cancer patients [7,8]. Intriguingly, in docetaxel treated patients, ABI was also effective in delaying the development of Skeletal Related Events (SREs) and radiological skeletal progression [3]. In chemotherapy-naïve mCRPC men, ABI plus prednisone significantly improved clinical outcome in terms of radiographic progression-free survival and overall survival compared with placebo plus prednisone [4,5,6]. Although, this trial did not include bone-related endpoints, the benefit of ABI in preventing SREs and bone disease progression was subsequently confirmed in this setting [9,10]. In particular, a recent ancillary analysis of the Italian real-world study ABITUDE demonstrated that ABI is able to modulate bone metabolism in the metastatic microenvironment [10]. The authors found a reduction in the main bone turnover markers (BTMs) such as alkaline phosphatase (BALP) and type-I collagen-C-telopeptide (CTX-1) after ABI treatment in patients with bone metastases. Interesting, similar findings were observed in patients who had not undergone treatment with bone targeted agents, confirming that BTM reduction was due to ABI treatment [10]. These data are in accordance with our previous findings that showed a significant decrease in BTMs in serum levels in ABI-treated patients [11]. In this study, we also observed a direct effect of ABI on primary human osteoclasts and osteoblasts (OBs) in terms of differentiation and activity. In particular, ABI exerted a direct bone anabolic and anti-resorptive effect modulating the expression of the main osteoblastic and osteoclastic gene markers [11].

Overall, these evidences suggest that ABI exerts a key role in the bone tumor microenvironment targeting both tumor and bone cells and, potentially, interfering with tumor-–bone interactions. To elucidate its direct effect on the bone microenvironment, we performed a large-scale proteomic analysis to investigate the OB modulation induced by ABI treatment. We focused on OBs since several previous papers reported their direct involvement in prostate tumor growth through the release of soluble molecules that, in turn, modulate androgen dependent and independent proliferative signaling [12,13,14,15,16].

## 2. Materials and Methods

### 2.1. Prostate Cancer Cell Line

C4-2B cells, gifted by Dr. Thalman [17], were cultured in T-medium (80% DMEM (Corning, New York, NY, USA), 20% DMEM/F-12 (Corning), 3 g/L NaHCO_3_ (Sigma Aldrich, St. Louis, MO, USA), 1% penicillin/streptomycin (Euroclone, Milano, Italy), 5 mg/mL insulin (Sigma Aldrich), 13.6 pg/mL triiodothyronine (Sigma Aldrich), 5 mg/mL apo-transferrin (Sigma Aldrich), 0.25 mg/mL biotin (Sigma Aldrich), and 25 mg/mL adenine (Sigma Aldrich)) with 10% of charcoal stripped serum (Sigma Aldrich). C4-2B Firefly/Renilla (C4-2B FR) cells were obtained using Cignal Androgen Receptor (AR) luciferase reporter assay (Qiagen, Hilden, Germany). Briefly, the cells were transfected with two lentiviral particles, one expressing a transcription factor-responsive reporter gene (firefly luciferase) under the control of the AR promoter element and the other expressed Renilla luciferase under the control of a housekeeping promoter. Firefly and Renilla positive cells were selected by adding 100 mg/mL of Hygromycin (Sigma Aldrich) and 2 mg/mL of Puromycin (Sigma Aldrich).

PC-3 cells, purchased from ATCC, were grown in RPMI-1640 medium (Corning), 10% of charcoal stripped serum (Sigma Aldrich), 2 mM L-glutamine (Euroclone) and 1% penicillin/streptomycin (Euroclone).

### 2.2. Human Osteoblasts

Primary OBs were generated from bone marrow mesenchymal stem cells isolated from male patients who underwent primary total hip replacement surgery at Fondazione Policlinico Universitario Campus Bio-Medico. Mesenchymal cells were differentiated in OBs in alpha MEM (Corning) supplemented with 15% of charcoal stripped serum (Sigma Aldrich), 2 mM L-glutamine (Euroclone), and 1% penicillin/streptomycin (Euroclone), 10 mM betaglycerophosphate (Sigma-Aldrich), 50 μM ascorbic acid (Sigma-Aldrich) and 100 nM dexamethasone (Sigma-Aldrich) [18]. During the differentiation protocol (from day 1 to 21), 10 μM of ABI or DMSO as control were added every three days. Alkaline phosphatase (ALP) and Alizarin red staining were performed to confirm that OB differentiation had occurred.

### 2.3. Osteoblast Conditioned Media

Osteoblast-conditioned media (OCM) was collected from OBs pre-treated with ABI (OCM ABI) or not (OCM CTRL) after 48 h of starvation in T-medium supplemented with 0.5% of charcoal stripped serum. For cell cycle analysis, OCM ABI or OCM CTRL was added to C4-2B FR or PC-3 cells seeded 6 × 10^4^ in 24-well plates for 96 h; for AR activity assay, OCM ABI or CTRL was administrated to C4-2B FR cells at a confluency of 10^4^ in 96-well plates for 24 h.

### 2.4. AR Activity Assay

AR activation status was determined using Dual-Luciferase Reporter Assay (Promega, Madison, WI, USA) according to the manufacturer’s instructions. Firefly and renilla luciferase signals were measured sequentially through a spectrofluorometer (Tecan Infinite M200Pro, Tecan, Milan, Italy). AR activity was quantified normalizing firefly luciferase signal with Renilla luciferase signal.

### 2.5. Cell Cycle Analysis

Cell cycle analysis was performed by flow cytometry as previously described (16). Briefly, cells were stained with Fixable Viability Dye conjugated with eFluor780 fluorochrome (eBioscience-Thermo Fisher Scientific, Waltham, MA, USA), fixed and permeabilized with Foxp3/Transcription Factor Staining Buffer Set (Thermofisher eBioscience, Waltham, MA, USA) and, then, incubated with anti-Ki67-APC antibody (clone 20Raj1 eBioscience) and Propidium Iodide (PI) solution (50 mg/mL PI + 40 ng/mL RNAseA + 0.1% of Triton) (Sigma Aldrich). FACS analysis were performed on CytoFlex instrument (Beckman Coulter, Brea, CA, USA) and data analyzed using CytExpert Software,v.2.1.

### 2.6. Proteomic Assay

The expression levels of 507 human proteins were analyzed in OCM ABI or OCM CTRL using the human L507 Array Membrane (RayBiotech, Peachtree Corners, GA, USA) following the manufacture’ instructions. In brief, the primary amines of samples were biotinylated and added on the membrane array. After over-night incubation, the HRP-Conjugated Streptavidin was added and, then, the signals were visualized by ChemiDoc MTP Imaging System (Bio-Rad, Milano, Italy). Band signals were quantified using ImageLab Software (Bio-Rad) and normalized using the positive membrane internal controls (anti-HRP and anti-streptavidin control signals).

### 2.7. Statistical Analysis

Graphs and statistical calculations were performed using GraphPad Prism (San Diego, CA, USA). Student’s t-test and one-way ANOVA test followed by Tukey’s multiple comparison tests were used to analyze the significance of the difference between samples. A *p* value of <0.05 was considered significant.

## 3. Results

### 3.1. ABI Modulate Osteoblast Secretome

To evaluate if ABI treatment influenced the secretion of OB soluble molecules, an extensive proteomic analysis was performed on OCM collected from OB treated (OCM ABI) or not with ABI (OCM CTRL). Out of 507-screened molecules, 180 factors were detected in OCM and 39 of them were differentially expressed in OCM ABI compared to OCM CTRL (*p* < 0.05). Pathway enrichment analysis revealed that a cluster of these differentially expressed molecules is positively associated with cell proliferation pathways [false discovery rate (FDR) adjusted *p*-value = 0.0019] (Figure 1).

This cluster included the insulin-like growth factor-binding proteins (IGFBP) 2, 4 and 6, low-density lipoprotein receptor-related protein-6 (LRP6), neuregulin 1 (NRG1), TIMP metallopeptidase inhibitor 1 (TIMP-1) and Vascular-Endothelial Growth Factor A (VEGF-A) and VEGF-C (Figure 2A). Interestingly, all these soluble factors were down-modulated by ABI, suggesting its potential anti-tumor effect mediated by OBs.

### 3.2. ABI Exerts an “Indirect” Anti-Tumor Effect Mediated by Osteoblasts

Next, to investigate if the OB modulation by ABI effectively influenced prostate cancer proliferation, we performed a functional in vitro validation using C4-2B cells, the best-characterized CRPC cell model (17). In particular, cell cycle analysis was carried out on C4-2B FR cells treated with OCM ABI or OCM CTRL. Data showed that OCM ABI significantly reduced the percentage of Ki-67 positive cells (*p* = 0.022) confirming the “indirect” anti-tumor effect of ABI mediated by OBs. To explore if this anti-tumor effect was AR-dependent, we analyzed AR activation status of C4-2B FR cells treated with OCM ABI or OCM CTRL. We found a significant reduction in AR activity in C4-2B FR cells after OCM ABI treatment (*p* = 0.017) suggesting that ABI inhibited the secretion of OB molecules involved in AR-dependent prostate cancer cell proliferation (Figure 2B). To investigate this hypothesis, androgen negative PC-3 cells were treated with OCM ABI or OCM CTRL. Data showed that OCM ABI did not affect PC-3 cell proliferation (Figure 2C) suggesting that the indirect anti-tumor effect of ABI was AR-mediated.

## 4. Discussion

About 70% of advanced prostate cancer patients develop bone metastases [19] usually associated with SREs including pathological fractures, palliative radiation to the bone, orthopedic surgery to prevent or repair a fracture, spinal cord compression and hypercalcemia. SREs are often correlated with severe pain, loss of mobility, reduced overall survival, decreased quality of life and increased health care costs [20]. Bone metastases from prostate cancer are predominantly characterized by osteoblastic lesions with a concurrent osteolytic component. Bone microenvironment represents a fertile soil where the reciprocal interplay between bone and cancer cells stimulate tumor growth and progression described as the “vicious cycle” [21,22,23,24]. The main finding of this study is that an anti-androgen agent, specifically designed to target cancer cells, shows the ability to directly modulate bone microenvironment and, in particular OB functions, enhancing its anti-tumor effect. Although the crucial role of OB signaling in prostate tumor proliferation is widely demonstrated, how OBs support tumor growth is not fully clarified. Data from previous papers suggest that multiple OB signals promote prostate cancer growth through androgen-dependent and independent mechanisms [12,13,14,15,16].

Our study reveals that ABI is able to inhibit some of these proliferative signals exerting an “indirect” anti-tumor effect mediated by OBs. We observed a reduced proliferation in C4-2B cells, but not in AR-negative PC-3 cells, suggesting that this effect was AR-dependent. In particular, ABI treatment reduced the OB release of IGFBP2, 4 and 6 proteins, known to be directly involved in prostate cancer progression via IGF-dependent and independent mechanisms [25,26]. Indeed, numerous evidences support a reciprocal cross talk between AR and IGF signaling that enhances AR transactivation and prostate cancer growth [27]. In particular, IGFB2 exerts a pivotal role in prostate cancer progression through IGF signaling or by PTEN phosphorylation and PI3K/Akt pathway activation [28,29,30]. Few evidences are reported about the role of IGFBP4 and 6 on prostate cancer proliferation [31,32]. Another soluble molecule inhibited by ABI is LRP6, an essential Wnt co-receptor for Wnt/β-catenin signaling. It has been demonstrated that the crosstalk between Wnt/β-catenin and AR signaling in CRPC leads to the abnormal expression of genes involved in cell proliferation [33,34]. In addition, ABI treatment down-modulated NRG1, recently identified in prostate tumor microenvironment as directly involved in antiandrogen resistance [35]. Zhang et al. found that NRG1 secreted by stromal cells in tumor milieu promote antiandrogen resistance activating HER3 signaling in prostate cancer cells and the blockade of the NRG1/HER3 axis can re-sensitize cancer cells to antiandrogen therapy [35]. Moreover, we found a significant reduction in VEGF-A and VEGF-C secretion by ABI suggesting and confirming the importance of VEGF signaling in prostate cancer progression. Intriguingly, a recent study demonstrated that VEGF-C contributed to prostate cancer cell proliferation by binding its specific receptor VEGFR-3 [36]. The authors found that targeting VEGFR-3 using a specific inhibitor reduced prostate cancer cell proliferation in vitro and blocked the tumor growth in xenograft mouse models [36]. In addition, VEGF-C is able to increase directly AR co-activators expression inducing AR transactivation [37] suggesting that its proliferative effect is AR-mediated. Finally, we observed a reduction in TIMP-1, which was able to promote tumor growth independently of its MMP inhibitory activity. In particular, several evidences reported the ability of TIMP-1 to protect cancer cells from apoptosis and induce epithelial-to-mesenchymal transition by binding its receptor CD63 [38,39,40,41]. In mCRPC patients, TIMP-1 is a negative prognostic biomarker associated with decreased survival [42].

Taken together, these data suggest that ABI could reprogram OB secretome creating a less permissive microenvironment for prostate cancer growth. This study provides the first evidence of an indirect antitumor effect of ABI through the modulation of multiple OB proliferative signals.

In the present study, we did not identify a specific factor or pathway responsible for the antitumor effect ABI-mediated. Although it could seem a limitation, we suppose that the inhibition of multiple OB molecules, rather than a single one, by ABI may result in a reduced cell proliferation. The use of OB conditioned media and prostate cancer cell lines might be considered as a too simplified and limited biological system to reproduce the complexity of bone/prostate cancer interactions. However, the primary human source of OBs obtained from different donors provides a model of “physiological” bone that retains the morphological and functional characteristics of their origin and capture the intra-individual heterogeneity. Moreover, the castration resistant C4-2B cells and the AR-negative PC-3 cells represent the most characterized bone metastatic prostate cancer models [17,43].

In conclusion, this explorative study provides biological information about ABI activity with a potential clinical utility. Indeed, our extensive proteomic analysis allows us to elucidate the effect of ABI on the modulation of bone/cancer crosstalk and identify novel potential therapeutic targets in castration resistant bone metastatic prostate cancer.

## Figures and Tables

**Figure 1 biomedicines-10-02154-f001:**
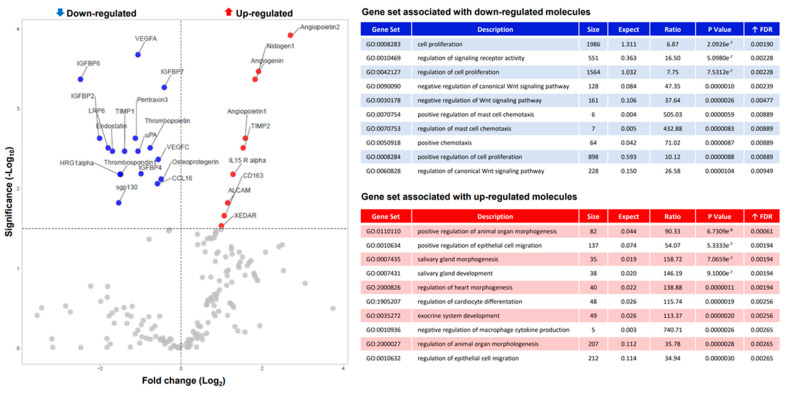
Volcano Plot of down (blue) and up-regulated soluble molecules (red) in OCM ABI. Top 10 gene SET ranked by enrichment score associated with down and up-regulated soluble molecules.

**Figure 2 biomedicines-10-02154-f002:**
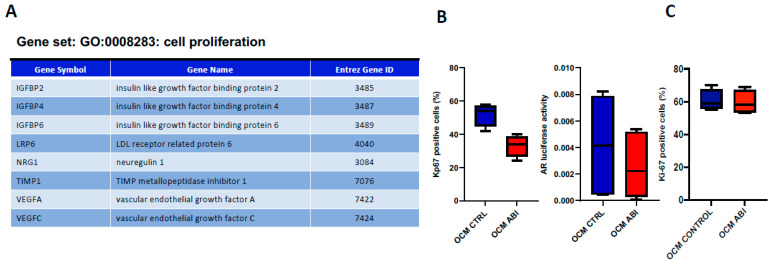
(**A**). Soluble molecules down-modulated in OCM ABI and associated with gene set “GO:0008283: cell proliferation” (**B**). Proliferation analysis and AR activation status in C4-2B FR treated with OCM ABI or OCM CTRL. (**C**) Proliferation analysis in PC-3 cells treated with OCM ABI or OCM CTRL.

## Data Availability

All data generated or analyzed during this study are included in the article.

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
