# Peer review of "Osteoblast Secretome Modulated by Abiraterone Treatment Affects Castration Resistant Prostate Cancer Cell Proliferation"

_biomedicines, 2022, doi:10.3390/biomedicines10092154_

Round 1

Reviewer 1 Report

This is a very well written manuscript providing further data regarding the complex interaction between prostate cancer and osteoblasts/bone microenvironment.  The authors additional demonstrations of the potential direct of effect on abiraterone on osteoblasts is intriguing.

The limitations include that only one prostate cancer cell line was used. The comparison to an alternative cell, perhaps a castrate sensitive or cell line derived from visceral or nodal (non bone metastases) would provide addition information.   Another limitation is that it does not contribute to a more specific pathway knowledge of the osteoblast-prostate cancer cell interaction and the actual amount of data/results is really quite limited.

Author Response

Reply: as suggested by reviewer, we provided to perform in vitro validation using another prostate cancer cell model. Unfortunately, we did not have the availability of castrate sensitive or cell line derived from visceral or nodal metastases, but only PC-3 cells. Despite PC-3 derives from bone metastases, there are androgen receptor (AR)-negative and provide useful information about the role of AR in osteoblast-mediated proliferation. Intriguingly, differently from C4-2B, the adding of osteoblast conditioned media (OCM), pre-treated with abiraterone, did not affect PC-3 proliferation. We provided to include these results in “result section” and figure 2C.

As underlined by reviewer we did not identified a specific pathway and this seems to be a limitation. However, we suppose that the inhibition of multiple osteoblast molecules, rather than a single one, by ABI are responsible of its anti-tumor effect mediated by osteoblasts. We discuss this point in “discussion section”. Our purpose was to submit a preliminary brief report in accordance with journal guidelines.

Reviewer 2 Report

The paper describes a proteomic analysis in order to elucidate the effect of ABI on osteoblast function and how this relation could affect prostate cancer progression. The paper in my opinion is well written and its topic sounds definitely interesting. Some issues have to be addressed: too many acronyms are used that make the paper difficult to read and follow. Please try to avoid this abundant use of acronyms also in the abstract. Regarding the results of the proteomic analysis made by the authors, I find it too preliminary and although the important future perspective of this study, I think the results are not adequately supported by experiments. They sound too hypothetical to me and only a cell cycle analysis has been reported by the authors. Please add more detailed experiments proving your conclusions in view also of the fact that no supplementary data are available. For these reasons I cannot recommend the publication of the manuscript in this current form.

Author Response

Reply: we agree with the reviewer that data are preliminary and, for this reason, our purpose was to submit a “brief report”. As reported in journal guidelines, this article type include “short studies that report preliminary results and, usually, contain two figures”. Regarding the experimental section, besides to cell cycle, we performed also the evaluation of androgen receptor activation status. Moreover, we provided to include additional experiments using androgen receptor –negative PC-3 cell line. Intriguingly, differently from C4-2B, the adding of osteoblast conditioned media (OCM), pre-treated with abiraterone, did not affect PC-3 proliferation suggesting that osteoblast mediated anti-tumoral effect of abiraterone was androgen receptor-dependent.  However, we included a detailed description of the study limitations in discussion section. Finally, we provided to reduce the use of acronyms mainly in the abstract section.

Reviewer 3 Report

In this study the authors analyze the effects of osteoblast (pretreated with abiraterone)  conditioned medium (OCM), on proliferation (KI67) and AR-signaling (reporter gene assay) in C4-2B prostate cancer cell lines. The authors suggest that abiraterone is able to modulate/reprogramm  osteoblasts secretome, thereby creating a less permissive microenvironment for prostate cancer.

MAJOR COMMENTS

(1) Human primary osteoblasts were obtained from healthy patients undergoing hip replacement. Did this group include only male donors?

 (2) The anti-proliferative effects were only tested in AR-positive castration resistant  LNCaP sublines  (C4-2B) isolated from bone metastasis. Does OCM conditioned media also affect KI67 levels in AR-negative bone metastatic PC-3 cells ? This experiment can easily be done and is absolutely necessary.

Author Response

Reply 1: osteoblasts were obtained all from male donors. We provided to specify this point in methods section

Reply 2: as suggested by the reviewer, we performed additional experiments using androgen receptor –negative PC-3 cell line. Differently from C4-2B,  we found that the adding of osteoblast conditioned media (OCM), pre-treated with abiraterone, did not affect PC-3 proliferation suggesting that osteoblast mediated anti-tumoral effect of abiraterone was androgen receptor-dependent.

Round 2

Reviewer 2 Report

I think the author introduce satisfactory changes to make the paper acceptable as a brief article. For this reason I recommend it’s publication in this current form.